# The cooperative IGS RT-GIMs: a reliable estimation of the global ionospheric electron content distribution in real-time

Qi Liu[1], Manuel Hernández-Pajares[1,2], Heng Yang[3,1], Enric Monte-Moreno[4], David Roma-Dollase[2], Alberto García-Rigo[1,2], Zishen Li[5], Ningbo Wang[5], Denis Laurichesse[6], Alexis Blot[6], Qile Zhao[7,8], Qiang Zhang[7], André Hauschild[9], Loukis Agrotis[10], Martin Schmitz[11], Gerhard Wübbena[11], Andrea Stürze[12], Andrzej Krankowski[13], Stefan Schaer[14,15], Joachim Feltens[16], Attila Komjathy[17], and Reza Ghoddousi-Fard[18]

[1]Universitat Politècnica de Catalunya (UPC-IonSAT), Barcelona, Spain.
[2]Institut d'Estudis Espacials de Catalunya (IEEC), Barcelona, Spain.
[3]School of Electronic Information and Engineering, Yangtze Normal University, 408100 Chongqing, China.
[4]Department of Signal Theory and Communications, TALP, Universitat Politècnica de Catalunya, 08034 Barcelona, Spain.
[5]Aerospace Information Research Institute (AIR), Chinese Academy of Sciences (CAS), Beijing, China.
[6]Centre National d'Etudes Spatiales, Toulouse, France.
[7]GNSS Research Center, Wuhan University, No. 129 Luoyu Road, Wuhan 430079, China.
[8]Collaborative Innovation Center of Earth and Space Science, Wuhan University, No. 129 Luoyu Road, Wuhan 430079, China.
[9]German Aerospace Center (DLR), German Space Operations Center (GSOC), 82234 Wessling, Germany.
[10]European Space Operations Center, European Space Agency, Darmstadt, Germany.
[11]Geo++ GmbH, Steinriede 8, 30827 Garbsen, Germany.
[12]BKG, Federal Agency for Cartography and Geodesy, Frankfurt, Germany.
[13]Space Radio-Diagnostics Research Centre, University of Warmia and Mazury in Olsztyn, 10-719 Olsztyn, Poland.
[14]Astronomical Institute of the University of Bern, Sidlerstrasse 5, Bern 3012, Switzerland.
[15]Federal Office of Topography (swisstopo), Wabern, Switzerland.
[16]Navigation Support Office, Telespazio Germany GmbH c/o European Space Agency/European Space Operations Centre, Robert-Bosch-Strasse 5, 64293 Darmstadt, Germany
[17]Near Earth Tracking Systems Group (335S), NASA - Jet Propulsion Laboratory, California Institute of Technology, 4800 Oak Grove Drive, M/S 138-317, Pasadena, CA 91109, USA.
[18]Canadian Geodetic Survey, Natural Resources Canada, Ottawa, Canada.

**Correspondence:** Manuel Hernández-Pajares (manuel.hernandez@upc.edu)

**Abstract.** The Real-Time Working Group (RTWG) of the International GNSS Service (IGS) is dedicated to providing high-quality data, high-accuracy products for Global Navigation Satellite System (GNSS) positioning, navigation, timing, and Earth observations. As one part of real-time products, the IGS combined Real-Time Global Ionosphere Map (RT-GIM) has been generated by the real-time weighting of the RT-GIMs from IGS real-time ionosphere centers including the Chinese Academy of Sciences (CAS), Centre National d'Etudes Spatiales (CNES), Universitat Politècnica de Catalunya (UPC), and Wuhan University (WHU). The performance of global Vertical Total Electron Content (VTEC) representation in all of the RT-GIMs has been assessed by VTEC from Jason3-altimeter during three months over oceans and dSTEC-GPS technique with 2-day observations over continental regions. According to the Jason3-VTEC and dSTEC-GPS assessment, the real-time weighting technique is sensitive to the accuracy of RT-GIMs. Compared with the performance of post-processed rapid Global Ionosphere Maps

(GIMs) and IGS combined final GIM (igsg) during the testing period, the accuracy of UPC RT-GIM (after the improvement of interpolation technique) and IGS combined RT-GIM (IRTG) is equivalent to the rapid GIMs and reaches around 2.7 and 3.0 TECU (TEC Unit, $10^{16}$el/m$^2$) over oceans and continental regions, respectively. The accuracy of CAS RT-GIM and CNES RT-GIM is slightly worse than the rapid GIMs, while WHU RT-GIM requires a further upgrade to obtain similar performance. In addition, the strong response to the recent geomagnetic storms has been found in the Global Electron Content (GEC) of IGS RT-GIMs (especially UPC RT-GIM and IGS combined RT-GIM). The IGS RT-GIMs turn out to be reliable sources of real-time global VTEC information and have great potential for real-time applications including range error correction for transionospheric radio signals, the monitoring of space weather and detection of natural hazards on a global scale. All the IGS combined RT-GIMs generated and analyzed during the testing period are available at http://doi.org/10.5281/zenodo.5042622 (Liu et al., 2021b).

## 1 Introduction

The Global Ionosphere Maps (GIMs), containing Vertical Total Electron Content (VTEC) information at given grid points (typically with a spatial resolution of 2.5 degrees in latitude and 5 degrees in longitude) have been widely used in both scientific and technological communities (Hernández-Pajares et al., 2009). Due to the high quality and global distribution of VTEC estimation, GIM has been applied to investigating the behavior of the ionosphere, such as the climatology of mean Total Electron Content (TEC), ionospheric anomalies before earthquakes, semiannual variations of TEC in ionosphere, the VTEC structure of polar ionosphere under different cases, W-index for ionospheric disturbance warning (e.g., Liu et al., 2009, 2006; Zhao et al., 2007; Jiang et al., 2019; Hernández-Pajares et al., 2020; Gulyaeva and Stanislawska, 2008; Gulyaeva et al., 2013). In addition, the high accuracy of GIM enables precise range corrections for transionospheric radio signals including radar altimetry, radio telescopes, and Global Navigation Satellite System (GNSS) positioning (e.g., Komjathy and Born, 1999; Fernandes et al., 2014; Sotomayor-Beltran et al., 2013; Le and Tiberius, 2007; Zhang et al., 2013a; Lou et al., 2016; Chen et al., 2020). The Center for Orbit Determination in Europe (CODE), European Space Agency (ESA), Jet Propulsion Laboratory (JPL), and Universitat Politècnica de Catalunya (UPC) agreed on the computation of individual GIMs in IONosphere map EXchange (IONEX) format, and created the Ionosphere Working Group (Iono-WG) of the International GNSS Service (IGS) at 1998 (Schaer et al., 1996, 1998; Feltens and Schaer, 1998; Feltens, 2007; Mannucci et al., 1998; Hernández-Pajares et al., 1998, 1999). In the IGS 2015 workshop, Chinese Academy of Sciences (CAS), Canadian Geodetic Survey of Natural Resources Canada (NRCan) and Wuhan University (WHU), became new Ionospheric Associate Analysis Centers (IAACs) (Li et al., 2015; Ghoddousi-Fard, 2014; Zhang et al., 2013b). Currently, there are three types of post-processed IGS GIMs at different latencies: final, rapid, and predicted GIMs. With the contribution from different IAACs, the final and rapid GIMs are assessed and combined by corresponding weights and uploaded to File Transfer Protocol (FTP) or Hypertext Transfer Protocol (HTTP) servers with the latency of 1-2 weeks and 1-2 days, respectively. The 1-day and 2-day predicted GIMs can provide valuable VTEC information in advance for ionospheric activities and corrections. However, the accuracy of predicted GIMs is limited due to the nonlinear variation of ionosphere and the lack of real-time ionospheric observations (Hernández-Pajares et al., 2009; García-Rigo et al.,

2011; Li et al., 2018).

In order to satisfy the growing demand for real-time GNSS positioning and applications, Real-Time Working Group (RTWG)
of IGS was established in 2001 and officially started to provide Real Time Service (RTS) in 2013 (Caissy et al., 2012; El-sobeiey and Al-Harbi, 2016). Aside from multi-GNSS real-time data streams, the IGS-RTS also generates RT-GNSS products streams, including satellite orbits, clocks, code/phase biases, and GIM. These high-quality IGS-RTS products enable precise GNSS positioning, navigation, timing (PNT), ionosphere monitoring, and hazard detection. In Radio Technical Commission for Maritime Services (RTCM) Special Committee (SC-104), the State Space Representation (SSR) correction data format is
defined as the standard message (RTCM-SSR) for real-time GNSS applications. In support of flexible multi-GNSS applications within current multi-constellation and multi-frequency environments, a new format (IGS-SSR) is developed. The dissemina-tion of IGS Real-Time Global Ionosphere Maps (RT-GIMs) adopts spherical harmonic expansion to save the bandwidth in both RTCM-SSR and IGS-SSR format (RTCM-SC, 2014; IGS, 2020).

The accuracy of RT-GIMs is typically worse than post-processed GIMs due to the short span of ionospheric observations,
sparse distribution of stations, higher noises in carrier-to-code leveling, or difficulty in carrier ambiguity estimation in real-time processing mode. While RT-GIMs perform slightly worse than post-processed GIMs, it is found that RT-GIMs are helpful to reduce the convergence time of dual-frequency Precise Point Positioning (PPP) and strengthen the solution (Li et al., 2013). With the corrections of RT-GIMs, the accuracy of single-frequency PPP reaches decimeter and meter level in horizontal and vertical directions (Ren et al., 2019), while the instantaneous (single-epoch) Real-Time Kinematic (RTK) Positioning over
medium and long-baseline is able to obtain higher success rate of the ambiguity fixing and reliability for rover stations in few centimeters level (Tomaszewski et al., 2020). In addition, the feasibility of ionospheric storm monitoring based on RT-GIMs is tested (García Rigo et al., 2017). A first fusion of IGS-GIMs and ionosondes data from the Global Ionosphere Radio Observa-tory (GIRO) paves the way for the improvement of real-time International Reference Ionosphere (Froń et al., 2020). Currently, the routine RT-GIMs are available from CAS, Centre National d'Etudes Spatiales (CNES), German Aerospace Center in Neu-
trelitz (DLR-NZ), JPL, UPC, WHU, and IONOLAB (Li et al., 2020; Laurichesse and Blot, 2015; Jakowski et al., 2011; Hoque et al., 2019; Komjathy et al., 2012; Roma Dollase et al., 2015; Sezen et al., 2013). Individual RT-GIMs from different IGS centers can be gathered from IGS-RTS by means of Network Transportation of RTCM by Internet Protocol (NTRIP) (Weber et al., 2007). With the contribution of IGS RT-GIMs from CAS, CNES and UPC, a first IGS real-time combination of GIMs was generated in 2018 (Roma-Dollase et al., 2018a).

Recently, one of the IGS RT-GIMs (UPC-IonSAT) has completely changed the real-time interpolation strategy, with a signifi-cant improvement. In addition, the number of contributing centers has been increased from 3 to 4, thanks to the participation of Wuhan University. A new version of IGS combined RT-GIM (IRTG) has been developed to improve the performance and also adapt to the newly updated IGS-SSR format. In addition, the developed software has been further parallelized to decrease the latency of IRTG computation to a few minutes (Tange, 2011). This paper summarizes the computation methods of IGS
RT-GIMs from different ionosphere centers and the generation of IRTG. In addition, the performance of different RT-GIMs and real-time weighting technique is shown and discussed. The conclusions and future improvements are given in the final section.

## 2 Data and methods

### 2.1 Real-time GNSS data processing

In order to generate RT-GIMs, the real-time GNSS observations from worldwide stations are received and transformed into Slant TEC (STEC). It should be noted that extraction of STEC in an unbiased way can be obtained by fitting an ionospheric model to the observations. With the global distributed STEC, different strategies are chosen for the computation of RT-GIMs. Currently, two methods are commonly used for the calculation of real-time STEC. The first method is the so-called Carrier-to-Code Levelling (CCL) as shown in Eq. 3 (Ciraolo et al., 2007; Zhang et al., 2019). The geometry-free (GF) combination of pseudorange and carrier phase observations is formed to extract STEC and DCB in an unbiased way by fitting an ionospheric model (for example, spherical harmonic model). Due to the typically shorter phase-arc length in real-time mode, the impact of multipath and thermal noise is higher than in post-processing mode (Li et al., 2020).

$$P_{GF,t} \equiv P_{2,t} - P_{1,t} = \alpha_{GF} \cdot STEC_t + c \cdot (D_r + D^s) + \epsilon_M + \epsilon_{\mathrm{T}} \tag{1}$$

$$L_{GF,t} \equiv L_{1,t} - L_{2,t} = \alpha_{GF} \cdot STEC_t + B_{GF} \tag{2}$$

$$\tilde{P}_{GF,t} \equiv L_{GF,t} - \frac{1}{k} \sum_{i=1}^{k} (L_{GF,i} - P_{GF,i}) \approx \alpha_{GF} \cdot STEC_t + c \cdot (D_r + D^s)) \tag{3}$$

where $P_{1,t}$ and $P_{2,t}$ are the pseudorange observations of epoch $t$ at first and second frequency, respectively. $\alpha_{GF}$ can be approximated as $40.3 \left( \frac{1}{f_2^2} - \frac{1}{f_1^2} \right)$. $f_1$ and $f_2$ are the first and second frequency of observation. $STEC_t$ is the STEC of epoch $t$. $r$ is receiver and $s$ is satellite. $c$ is the speed of light in vacuum. $D_r$ and $D^s$ are the receiver Differential Code Biases (DCB) and satellite DCB. $\epsilon_M$ and $\epsilon_T$ are the code multipath error and thermal noise error. $L_{1,t}$ and $L_{2,t}$ are the carrier phase observations including the priori corrections (such as wind-up term) of epoch $t$ at first and second frequency. $B_{GF}$ equals to $B_1 - B_2$, while $B_1$ and $B_2$ are the carrier phase ambiguities including the corresponding phase bias at first and second frequency, respectively. $k$ is the length of smoothing arc from beginning epoch to epoch $t$, and $\tilde{P}_{GF,t}$ represents the smoothed $P_{GF}$ of epoch $t$ which is significantly affected by the pseudorange multipath in real-time mode than in post-processing.

The second method is the GF combination of phase-only observations, and the $B_{GF}$ is estimated together with the real-time TEC model (for example, described in terms of tomographic voxel-based basis functions) in Eq. 2 (Hernández-Pajares et al., 1997, 1999). Although the STEC from the second method is accurate and free of code multipath and thermal noise in post-processing, the convergence time can affect the accuracy of the STEC, most likely in the isolated receivers. In addition, the computation methods of RT-GIMs from different IGS real-time ionosphere centers were compared in detail at the next subsection and summarized in Table 1. Some ionosphere centers (CAS, CNES, WHU) directly estimate and disseminate spherical harmonic coefficients in sun-fixed reference frame as Eq. 4 (RTCM-SC, 2014; Li et al., 2020), while UPC generates

**Table 1.** The brief summary of different IGS RT-GIMs

| Agency /GIM | Runing date | Extra ionospheric information | DCB computation | GIM computation |
|---|---|---|---|---|
| CAS | Mid-2017 to present | 2-day predicted GIM as background information | Estimated at the same time with local VTEC, and corrected by three-day aligned code bias | Observations with predicted GIMs generate 15-degree spherical harmonic expansion GIM in solar-geographic frame |
| CNES | End-2014 to present (with an evolution of the spherical harmonic degree) | No | Expected in a forthcoming version | 12-degree spherical harmonic expansion GIM which is generated in solar-geographic frame |
| UPC /URTG | 2011-02-06 to 2019-09-08 | 1-2 day rapid GIM UQRG as background information | optional | Tomographic model with kriging interpolation method and frozen rapid GIM (UQRG) as a priori model, which generates RT-GIM in sun-fixed geomagnetic frame |
| UPC /USRG | 2019-09-08 to present | 1-2 day rapid GIM UQRG as background information | optional | Tomographic model with spherical harmonic interpolation method and frozen rapid GIM (UQRG) as a priori model, which generates RT-GIM in sun-fixed geomagnetic frame |
| UPC /UADG | 2021-01-04 to present | historical UQRG (since 1996) as databases | optional | Tomographic model adopting atomic decomposition and LASSO solution for the global interpolation with the help of historical GIMs (UQRG), which generates GIM in sun-fixed geomagnetic frame |
| WHU | 2020-11-09 to present | 2-day predicted GIM as background information | Directly use the previous satellite and receiver DCB estimated simultaneously with WHU rapid GIM | Observations with predicted GIMs yield 15-degree spherical harmonic expansion GIM in solar-geomagnetic frame |

the RT-GIM in IONEX format and transforms RT-GIM into spherical harmonic coefficients for the dissemination.

$$
\begin{cases}
M_z = [1 - \sin^2 z/(1 + H_{ion}/R_E)^2]^{-\frac{1}{2}} \\
VTEC_t = STEC_t/M_z \\
VTEC_t = \sum_{n=0}^{N_{SH}} \sum_{m=0}^{\min(n,M_{SH})} P_{n,m}(\sin\varphi_I) \cdot (C_{n,m}\cos(m\lambda_{S,t}) + S_{n,m}\sin(m\lambda_{S,t})) \\
\lambda_{S,t} = (\lambda_I + (t - t_0) * \pi/43200) \text{ modulo } 2\pi
\end{cases}
\tag{4}
$$

where $z$ is the satellite zenith angle, $M_z$ is the mapping function between $STEC_t$ and $VTEC_t$. $H_{ion}$ is the height of iono-spheric single-layer assumption, and $R_E$ is the radius of the earth. $VTEC_t$ is the VTEC of epoch $t$. $N_{SH}$ is the max degree of spherical harmonic expansion, and $M_{SH}$ is the max order of spherical harmonic expansion. $n, m$ are corresponding indices. $P_{n,m}$ is the normalized associated Legendre functions. $C_{n,m}, S_{n,m}$ are sine and cosine spherical harmonic coefficients. $\varphi_I$ and $\lambda_I$ are the geocentric latitude and longitude of Ionospheric Pierce Point (IPP). $\lambda_{S,t}$ is the mean sun fixed and phase-shifted longitude of IPP of epoch $t$ (typically shifted by 2 hours to approximate TEC maximum at 14:00 in local time). $t$ is the current epoch. $t_0$ is a common reference of shifted hours, taken as 0 hours in the present broadcasting of RT-GIM for WHU and 2 hours for CAS, CNES, and UPC.

## 2.2 The computation of RT-GIMs by different IGS real-time ionosphere centers

The strategies for generating RT-GIMs differ between IGS real-time ionospheric analysis centers (ACs). In this subsection, a brief introduction on the generation of RT-GIMs from individual ACs as well as the strategy comparison between different ACs are given.

### 2.2.1 Chinese Academy of Sciences

The post-processed GIM of CAS has been computed and uploaded to IGS since 2015 (Li et al., 2015). A predicting-plus-modeling approach is used by CAS for the computation of RT-GIM (Li et al., 2020). CAS RT-GIM is generated with multi-GNSS, GPS and GLONASS L1+L2, BeiDou B1+B2 and Galileo E1+E5a real-time data streams, provided by the IGS and regional GNSS tracking network stations. The real-time differential code biases (DCB) are estimated as part of the local ionospheric VTEC modeling using a generalized trigonometric series (GTS) function as Eq. 5. And then three-day aligned biases are incorporated to increase the robustness of real-time DCBs (Wang et al., 2020).

$$
\begin{cases}
STEC_t = M_z \cdot VTEC_t + c \cdot (D^s + D_r) \\
VTEC_t = \sum_{i=0}^{i_{max}} \sum_{j=0}^{j_{max}} \left\{ E_{i,j} \cdot \varphi_d^i \cdot \lambda_d^j \right\} + \sum_{l=0}^{l_{max}} \left\{ C_l \cos(l \cdot h_t) + S_l \sin(l \cdot h_t) \right\} \\
h_t = 2\pi(t - 14)/T, \quad T = 24h \\
i_{max} = j_{max} = 2 \\
l_{max} = 4
\end{cases}
\tag{5}
$$

where $r$ is receiver and $s$ is satellite. $\varphi_d$ and $\lambda_d$ are the difference between IPP and station in latitude and longitude, respectively. $i, j, l$ represent the degrees in the polynomials model and Fourier series expansion. $E_{i,j}, C_l, S_l$ are unknown parameters.

The real-time STEC is computed by subtracting estimated DCB in Eq. 5 from $\tilde{P}_{GF,t}$ in Eq. 3 and then the STEC is converted into VTEC by means of mapping function. The real-time VTEC from 130 global stations is directly modeled in a solar-geographic reference frame as Eq. 4. To mitigate the impacts of the unstable real-time data streams, e.g. the sudden interruption of the data streams, CAS predicted TEC information is also included for RT-GIM computation. The broadcasted CAS RT-GIM is computed by the weighted combination of real-time VTEC spherical harmonic coefficients and predicted ionospheric information (Li et al., 2020).

### 2.2.2   Centre National d'Etudes Spatiales

In the framework of the RTS of the IGS, CNES computes global VTEC in real-time thanks to the CNES PPP-WIZARD project since 2014. The real-time VTEC is extracted by pseudorange and carrier phase GF combination as Eq. 3 with the help of mapping function. And the single-layer assumption in the mapping function adopts an altitude of 450 km above the Earth. CNES also use spherical harmonic model for global VTEC representation, and the equation is the same as Eq. 4. Spherical harmonic coefficients are computed by means of a Kalman Filter and simultaneous STEC from 100 stations of the real-time IGS network. CNES started to broadcast RT-GIM at the end of 2014 and changed spherical harmonic degrees from 6 to 12 in May of 2017 (Laurichesse and Blot, 2015).

### 2.2.3   Universitat Politècnica de Catalunya

UPC has been providing daily GIMs in IONEX-format to IGS since 1998 (Hernández-Pajares et al., 1998, 1999; Orús et al., 2005). In order to meet the demand of real-time GIM, UPC developed the Real-Time TOMographic IONosphere model software (RT-TOMION) and started to generate the UPC RT-GIM on February 6 of 2011. The phase-only GF combination as Eq. 2, is used for obtaining real-time STEC from around 260 stations, and a 4-D voxel-based tomographic ionosphere model is adopted for global electron content modeling. The ionosphere is divided into two layers in the tomographic model and the electron density of each voxel is estimated together with the ambiguity term $B_{GF}$ by means of a Kalman filter in the sun-fixed reference frame. The estimated electron density is condensed at a fixed effective height (450 km) for the generation of a single-layer VTEC map, and then the VTEC interpolation method is adopted in a sun-fixed geomagnetic reference frame for filling the data gap on a global scale.

From 2011 to 2019, the kriging technique is selected by UPC for real-time VTEC interpolation. And the spherical harmonic model has been adopted by UPC since September 08 of 2019. Recently, a new interpolation technique, denoted as Atomic Decomposition Interpolator of GIMs (ADIGIM), has been developed. Since the global ionospheric electron content mainly depends on the diurnal, seasonal, and solar variation, ADIGIM is computed by the weighted combination of good-quality historical GIMs (e.g. UQRG) with similar ionosphere conditions. The database of historical GIMs cover the last two solar cycles since 1998. The method for obtaining the weights of the linear combination of past maps is based on Eq. 6, which was first introduced in the problem of face recognition (Wright et al., 2008, 2010). While the face recognition is affected by the occlusions (such as glasses) in the face image, the reconstruction of GIM has problems in the regions that are not covered by GNSS-stations. The problems have to be taken into account when selecting the past maps for combination, and should not

introduce a bias. As shown in Eq. 6, the problem is solved by introducing $\ell_2$ norm and $\ell_1$ norm. The property of the atomic decomposition and the least absolute shrinkage and selection operator (LASSO) is that it can select a small set of past maps which are the most similar to the real-time measured VTEC at IPPs. The ADIGIM technique minimizes the difference between observed VTEC measurement and weighted VTEC from historical UQRG in similar ionosphere conditions. The underlying assumption is that the VTEC distribution over the areas not covered by the IPPs can be represented by the elements of the

historical library of UQRG (Yang et al., 2021). The UPC RT-GIM with the new technique is denoted as UADG and generated by Eq. 6. Due to the improvement provided by the UADG, the broadcasted UPC-GIM was changed from USRG to UADG on January 04 of 2021. In addition, the USRG and UADG are generated in real-time mode and saved in IONEX format at HTTP as shown in Table 1.

$$\begin{cases} VTEC_{I,t} \approx D_{g,I,t} \cdot \alpha_t \\ \tilde{\alpha}_t = \arg\min_{\alpha_t} \frac{1}{2} \|VTEC_{I,t} - D_{g,I,t} \cdot \alpha_t\|_{\ell_2} + \rho \|\alpha_t\|_{\ell_1} \\ G_t = D_t \tilde{\alpha}_t \end{cases} \tag{6}$$

where $VTEC_{I,t}$ is the observed VTEC at IPP of epoch $t$. It is assumed that $VTEC_{I,t}$ can be approximated by $D_{g,I,t}$ and $\alpha_t$, while $D_{g,I,t}$ is the VTEC extracted at IPP from historical databases of GIM $g$ (For UPC, the UQRG is used) and $\alpha_t$ is the unknown weight vector of each historical GIM at epoch $t$. $\tilde{\alpha}_t$ is the estimated weight vector of each selected UQRG at epoch $t$. The estimated weight vector $\tilde{\alpha}_t$ is obtained by LASSO regression method with loss function norm $\ell_2$ and regularization norm $\ell_1$. $\ell_2$ is the norm for minimizing the euclidean distance between observed VTEC measurements and historical UQRG

databases at epoch $t$. $\ell_1$ is the regularization norm for penalizing the approximation coefficients to limit the number of UQRG involved in the estimation and $\rho$ controls the sparsity of solution. $G_t$ is the generated UPC RT-GIM of epoch $t$ and is the weighted combination of historical UQRG. For mathematical convenience, each 2-D GIM is reformed as a 1-D vector (I.e., the columns are stacked along the meridian in order to create a vector of all the grid points of the map). This is justified because the measure of similarity is done over cells of 2.5×5.0 degrees in the maps, and therefore the underlying $\mathbb{R}^2$ (coordinate space of dimension 2) structure is not relevant for computing euclidean distances in $\ell_2$ norm. $D_t$ is the selected historical UQRG

databases with similar ionosphere conditions at epoch $t$.

### 2.2.4   Wuhan University

The daily rapid and final GIM products have been generated with WHU new software named GNSS Ionosphere Monitoring and Analysis Software (GIMAS) since 21 June 2018 (Zhang and Zhao, 2018). At the end of the year 2020, WHU has also

published a first RT-GIM product.

WHU uses the spherical harmonic expansion model and the formula is identical to Eq. 4. Currently, only the GPS real-time data streams from about 120 globally distributed IGS stations are used. The double frequency code and carrier phase observations with a cut-off angle of 10 degrees are used to gather precise geometry-free ionospheric data with the CCL method as Eq. 3 and ionospheric mapping function with the layer height of 450 km. In order to avoid the influence of satellite and receiver

DCB on ionospheric parameters estimation, WHU directly uses the previous estimated DCB from WHU rapid GIM product. According to previous experience, the real-time data is not enough to model the ionosphere precisely on a global scale with

spherical harmonic expansion technique. Considering the lack and the uneven distribution of the GPS-derived ionospheric data, 2-day predicted GIM as external ionospheric information is also incorporated. It is important to balance the weight between the real-time data and the background information. Both the RT-GIM quality and the root mean square (RMS) map are influenced by the weight (Zhang and Zhao, 2019).

In the year 2021, WHU is going to focus on how to further improve the accuracy of RT-GIM and update the computation method. The precise WHU RT-GIMs with multi-GNSS data and the application of WHU RT-GIM in the GNSS positioning as well as space physics domain, are expected as next steps.

## 2.3 The combination of IGS RT-GIMs

Thanks to the contribution of the initial IGS real-time ionosphere centers (CAS, CNES, and UPC) and globally distributed real-time GNSS stations, the first experimental IRTG was generated by means of Real-Time dSTEC (RT-dSTEC) weighting technique (normalized inverse of the squared RMS of RT-dSTEC error) in October 2018 (Roma-Dollase et al., 2018a; Li et al., 2020). Recently, WHU published the first WHU RT-GIM and UPC upgraded the real-time VTEC interpolation technique. A new version of IRTG has been developed and broadcasted since January 04 of 2021. The IGS combined RT-GIM is based on the weighted mean value of VTEC from different IGS centers as Eq. 7.

$$
\begin{cases}
VTEC_{IRTG,t} = \sum_{g=1}^{N_{AC}} (w_{g,t} \cdot VTEC_{g,t}) \\
w_{g,t} = I_{g,t} / \sum_{g=1}^{N_{AC}} (I_{g,t}) \\
I_{g,t} = 1/RMS_{\delta,g,t}^2 \\
RMS_{\delta,g,t} = \sqrt{\sum_{i=1}^{N_t} (\delta_{g,i})^2 / N_t}
\end{cases}
\tag{7}
$$

where $VTEC_{IRTG,t}$ is the VTEC of IGS combined RT-GIM at epoch $t$ and $VTEC_{g,t}$ is VTEC of RT-GIM $g$ from IGS center at epoch $t$. $N_{AC}$ is the number of IGS centers. $w_{g,t}$ is the weight of corresponding RT-GIM $g$ at epoch $t$ (the sum of $w_{g,t}$ at epoch $t$ is 1). $RMS_{\delta,g,t}$ is the root mean square of RT-dSTEC error at epoch $t$. $I_{g,t}$ is the inverse of the mean square of RT-dSTEC error at epoch $t$. $N_t$ is the number of RT-dSTEC observations from the beginning epoch to current epoch $t$. $\delta_{g,i}$ is the RT-dSTEC error of RT-GIM $g$ in the RT-dSTEC assessment.

In addition, the RT-dSTEC assessment is based on Root Mean Square (RMS) of the dSTEC error calculated by Eq. 8. In order to adapt to the real-time processing mode, the reference STEC ambiguous measurement $L_{GF,t_{ref}}$ is set to be the first elevation angle higher than $10°$ within continuous phase-arc to enable the RT-dSTEC calculation in the elevation-ascending arc.

$$
\delta_{g,t} = \frac{1}{\alpha_{GF}} ((L_{GF,t} - L_{GF,t_{ref}}) - (M_z \cdot VTEC_{g,t} - M_{z_{ref}} \cdot VTEC_{g,t_{ref}}))
\tag{8}
$$

where $\delta_{g,t}$ is the dSTEC error of GIM $g$ at epoch $t$. $t_{ref}$ is the epoch when reference elevation angle is stored. $M_z$ and $M_{z_{ref}}$ are the mapping function of zenith angle of epoch $t$ and zenith angle of reference epoch $t_{ref}$, respectively.

Due to the limited number of real-time stations, 25 common real-time stations that have been used by all the IGS real-time ionosphere centers are selected for allowing a fair RT-dSTEC assessment. The distribution can be seen as Fig. 1. Therefore, the RT-dSTEC is the measurement of "internal" post-fit residuals of RT-GIMs, and still sensitive to the accuracy of assessed GIMs.

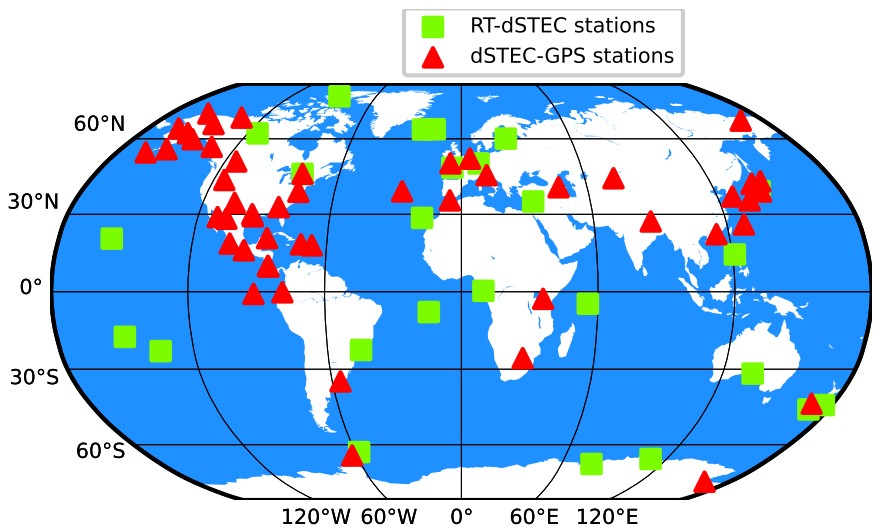

**Figure 1.** The 25 common real-time stations for RT-dSTEC assessment (in green color) and 50 external GNSS stations for dSTEC-GPS assessment (in red color)

Every 20 minutes, the RT-dSTEC assessment is performed and used for the combination of different IGS RT-GIMs. The steps for the generation of IRTG can be seen as Fig. 2. The RTCM-SSR has been the standard message for real-time corrections, and

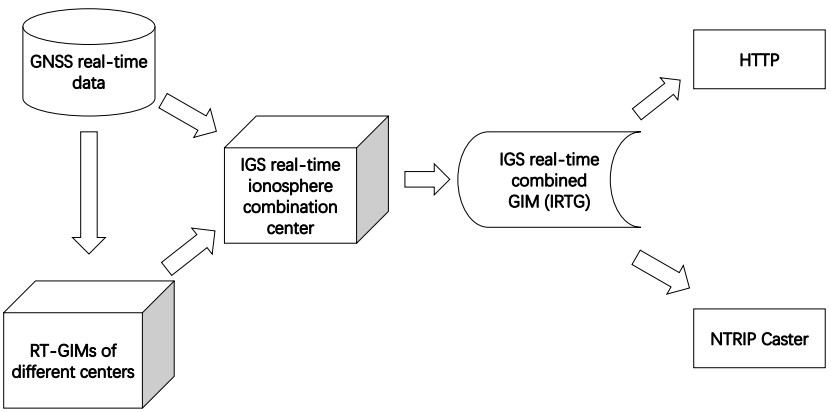

**Figure 2.** Data flow for the IGS real-time combined GIM

the IGS State Space Representation (SSR) Format Version 1.00 was published on October 05 of 2020 (IGS, 2020). The content of IGS-SSR is compatible with RTCM-SSR contents. And the IGS-SSR format can support more extensions such as satellite

attitude, phase center offsets and variations in the near future. At present, both RTCM-SSR and IGS-SSR formats are used for the dissemination of RT-GIMs. In addition, IGS defines different reference for antenna correction: Average Phase Center (APC), and Center of Mass (CoM). The current status of RT-GIMs from different ionosphere centers can be seen as Table 2. It should be noted that "SSRA" means the SSR with APC reference, and "SSRC" means the SSR with CoM reference.

**Table 2.** The current status of broacasting IGS RT-GIMs

| Agency | Temporal resolution | Broadcast frequency | Spherical harmonic degree | Mountpoints in NTRIP caster | Real-Time IONEX saved at FTP/HTTP |
|---|---|---|---|---|---|
| CAS | 5 minutes | 1 minute | 15 | 123.56.176.228:2101/CAS05[a]<br>59.110.42.14:2101/SSRA00CAS1[b]<br>59.110.42.14:2101/SSRA00CAS0[a]<br>59.110.42.14:2101/SSRC00CAS1[b]<br>59.110.42.14:2101/SSRC00CAS0[a]<br>182.92.166.182:2101/IONO00CAS1[b]<br>182.92.166.182:2101/IONO00CAS0[a] | ftp://ftp.gipp.org.cn/product/ionex/<br>(update at the end of day) |
| CNES | 2 minutes | 1 minute | 12 | products.igs-ip.net:2101/CLK91[a]<br>products.igs-ip.net:2101/SSRA00CNE1[b]<br>products.igs-ip.net:2101/SSRA00CNE0[a]<br>products.igs-ip.net:2101/SSRC00CNE1[b]<br>products.igs-ip.net:2101/SSRC00CNE0[a] | No |
| UPC (only UADG) | 15 minutes | 15 seconds | 15 | products.igs-ip.net:2101/IONO00UPC1[b] | http://chapman.upc.es/tomion/real-time/quick/<br>(UADG and USRG, update every 15 minutes) |
| WHU | 5 minutes | 1 minute | 15 | 58.49.58.150:2106/IONO00WHU0[a] | No |
| IGS | 20 minutes | 15 seconds | 15 | products.igs-ip.net:2101/IONO00IGS1[b] | http://chapman.upc.es/irtg/<br>(update every 20 minutes) |

[a] RTCM-SSR format

[b] IGS-SSR format

## 3   The performance of IGS RT-GIMs

In this section, the performance of IGS RT-GIMs was analyzed, and compared with IGS rapid GIMs as well as IGS combined final GIM. It should be noted that the RT-GIMs were gathered with BKG Ntrip Client (BNC) software (Weber et al., 2016) and

generated by received spherical harmonic coefficients from different centers as Table 2. And there were two kinds of temporal resolution for received RT-GIMs: the common temporal resolution of 20 minutes, and the full (original) temporal resolution. Since the IRTG is combined every 20 minutes, we will focus on such a common time resolution to compare the performance. The detail of compared RT-GIMs can be seen as Table 3. The influence of temporal resolution on RT-GIMs was also shown in this section.

Before detailing the JASON3-VTEC and GPS-dSTEC assessment, it should be taken into account that the GIM error versus JASON-VTEC measurements have a high correlation with the GIM error versus dSTEC-GPS based measurements, although the JASON-VTEC measurements are vertical and the dSTEC-GPS measurements are slant. As demonstrated in Hernández-Pajares et al. (2017), the Jason3-VTEC assessment and dSTEC-GPS assessment are independent and consistent for GIM evaluation. In other words, the slant ray path geometry changes does not affect the capability of dSTEC reference data to rank the GIM, and the electron content between the Jason3-altimeter and the GNSS satellites does not significantly affect the assessment of GIMs based on Jason3-VTEC data.

**Table 3.** The Id. of compared IGS RT-GIMs

| Agency | 20-minute RT-GIM | RT-GIM with full temporal resolution |
|--------|------------------|--------------------------------------|
| CAS | crtg | crfg |
| CNES | cnes | cnfs |
| UPC | upc1 | upf1 |
| WHU | whu0 | whf0 |
| IGS | irtg | irfg[a] |

[a] irfg and irtg are the same.

### 3.1 Jason3-VTEC assessment

The VTEC from the Jason3-altimeter was gathered as an external reference over the oceans. After applying a sliding window of 16 seconds to smooth the altimeter measurements, the typical standard deviation of Jason3-VTEC measurement error is around 1 TECU. Although the electron content above Jason3-altimeter (about 1300 km) is not available and the altimeter bias is around a few TECU, the standard deviation of the difference between GIM-VTEC and Jason3-VTEC is adopted to avoid the Jason3-altimeter bias and the constant bias component of the plasmaspheric electron content in the assessment. The plasmaspheric electron content variation is up to a few TECU and is relatively a small part when compared with the GIM errors over the oceans. And Jason3-VTEC has been proven to be a reliable reference of VTEC on the oceans where are the most challenging regions for GIMs (containing few nearby receivers in such regions) and typically far from permanent GNSS receivers potentially contributing to the GIM (Roma-Dollase et al., 2018b; Hernández-Pajares et al., 2017). In this context, the daily standard deviation of the difference between Jason3-VTEC and GIM-VTEC was suitable as the statistic for GIM

assessment in Eq. 9.

$$
\begin{cases}
Bias_g = \sum_{i=1}^{N_J}(VTEC_{Jason3,i} - VTEC_{g,i})/N_J \\
STD_g = \sqrt{\sum_{i=1}^{N_J}(VTEC_{Jason3,i} - VTEC_{g,i} - Bias_g)^2/(N_J - 1)}
\end{cases}
\tag{9}
$$

where $VTEC_{Jason,i}$ and $VTEC_{GIM,i}$ are VTEC extracted from Jason3 and GIM observation $i$, respectively. $N$ is the number of involved observations.

The recent three-month data (December 01 of 2020 to March 01 of 2021), containing the two significant events (new contribut-

ing RT-GIM (WHU) from January 03 of 2021 and the introduction of the new tomographic-atomic decomposition UPC-GIM (UADG) on January 04, 2021) has been selected to study the consistency and performance of the IGS RT-GIMs.

As can be seen in Fig. 3, the standard deviation of UPC RT-GIM (upc1) VTEC versus measured Jason3-VTEC is worse than other RT-GIMs before the transition from USRG to UADG on January 04, 2021. It should be noted that the upc1 in RTCM-SSR format was stopped from December 15 of 2020 to January 2 of 2021, due to the change of broadcasting format and some tech-

nical issues. The assessment of upc1 was based on the UPC RT-GIMs saved in local repository during the interrupted period. The standard deviation of upc1 VTEC versus measured Jason3-VTEC reached around 7 TECU on December 6 of 2020 due to the interruption of downloading module. And the upc1 achieved a significant improvement after the transition on January 04, 2021. In addition, the accuracy of IGS experimental combined RT-GIM (irtg) also increased due to the better performance of upc1. Compared with IGS rapid GIMs (corg, ehrg, emrg, esrg, igrg, jprg, uhrg, uprg, uqrg, whrg) and IGS final combined GIM

(igsg), the upc1 and irtg are equivalent to the post-processed GIMs and even better than some rapid GIMs. The accuracy of CAS RT-GIM (crtg) and CNES RT-GIM (cnes) are close to the post-processed GIMs, while WHU RT-GIM (whu0) is slightly worse than the other GIMs. As shown and explained in Eq. 4, the whu0 is shifted by 0 hours. To see the influence of phase-shifted $\lambda_{S,t}$, the whu0 is manually shifted by 2 hours (i.e., take $t_0$ as 2 hours for whu0 in Eq. 4) in post-processing mode. And the accuracy of the 2-hour shifted WHU RT-GIM (whu1) is slightly better than whu0 as can be seen in Fig. 3.

In order to investigate the influence of temporal resolution on RT-GIMs over oceans, different RT-GIMs with full temporal resolution were involved. The summary of Jason3-VTEC assessment can be seen in Table 4. The overall standard deviation of GIM-VTEC minus Jason3-VTEC is computed in separate time periods to focus on the influence of the transition from USRG to UADG. As shown in Table 4, the overall standard deviation of GIM-VTEC versus Jason3-VTEC is consistent with the Fig. 3 and the quality of 20-minute and full temporal resolution of RT-GIMs are similar over oceans. And the accuracy of 2-hour

shifted whu1 in Jason3-VTEC assessment is higher than whu0 in Table 4. In particular, the overall standard deviation of upc1 VTEC versus measured Jason3-VTEC drops from 4.3 to 2.7 TECU and, in agreement with that, the standard deviation of irtg VTEC versus measured Jason3-VTEC decreases from 3.3 to 2.8 TECU.

## 3.2 dSTEC-GPS assessment

In addition, dSTEC-GPS assessment in post-processing mode was involved as a complementary tool with high accuracy (better

than 0.1 TECU) over continental regions on a global scale. In the dSTEC-GPS assessment, the maximum elevation angle within a continuous arc was regarded as the reference angle in Eq. 8. The dSTEC observations provide the direct measurements of the

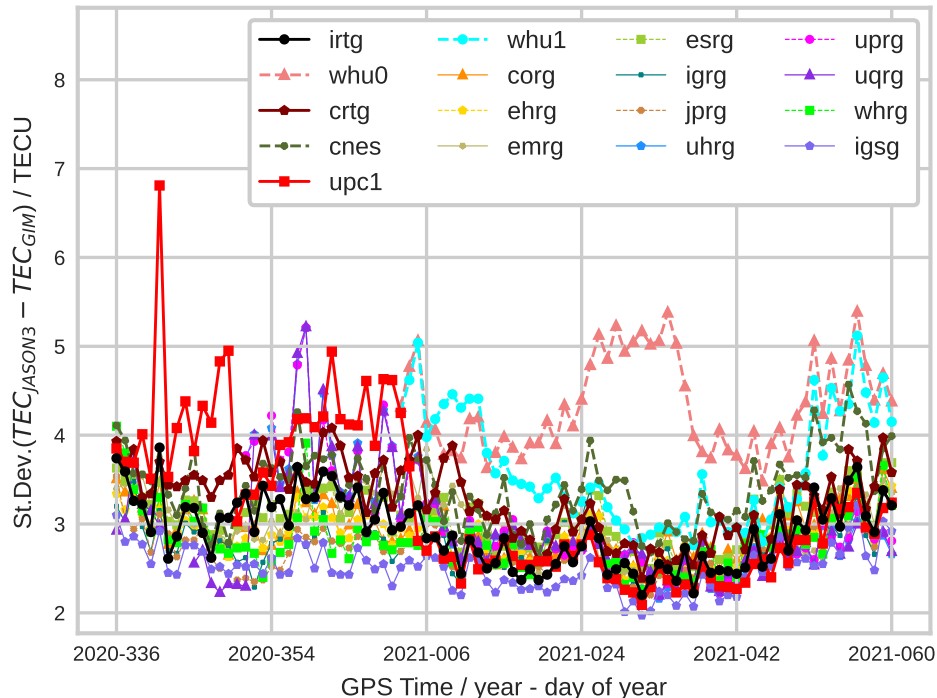

**Figure 3.** Daily standard deviation of GIM VTEC versus measured Jason3-VTEC (in TECU), from December 01 of 2020 to March 01 of 2021.

difference of STEC within a continuous phase-arc involving different geometries. And the mapping function is used by GNSS users to convert GIM-VTEC to GIM-STEC for GNSS positioning. Therefore, the dSTEC observations, containing different geometries and mapping function error, are accurate and direct measurements for evaluating ray path GIM-STEC which is commonly used by GNSS users to calculate ionospheric correction. In addition, the common agreed ionospheric thin layer model is set to be 450 km height in the generation of GIM to provide VTEC in a consistent way for different ionospheric analysis centers. And in this way the GNSS users are able to consistently recover a most accurate STEC from GIM-VTEC by the commonly agreed mapping function. The dSTEC-GPS assessment was performed by globally distributed GNSS stations as shown in Fig. 1 on January 03 (before the transition of UPC RT-GIM from USRG to UADG) and January 05 (after the transition) in 2021, with a focus on the transition of UPC RT-GIM. The RMS error and relative error were used for the assessment as Eq. 10.

$$
\begin{cases}
RMS_{\delta,g} = \sqrt{\sum_{i=1}^{N_S} (\delta_{g,i})^2 / N_S} \\
O_{\Delta_{S_{GPS}}} = (L_{GF,t} - L_{GF,t_{ref}})/\alpha_{GF} \\
RMS_{\Delta_{S_{GPS}}} = \sqrt{\sum_{i=1}^{N_S} (O_{\Delta_{S_{GPS}}})^2 / N_S} \\
Relative\ error_g = 100 \cdot RMS_{\delta,g}/RMS_{\Delta_{S_{GPS}}}
\end{cases}
\tag{10}
$$

**Table 4.** Standard deviation of GIM-VTEC minus Jason3-VTEC in Jason3-VTEC assessment (last two columns), and dSTEC-GPS assessment results of RT-GIMs on January 03 (second and third column) and January 05 (fourth and fifth column) in 2021.

| GIM | RMS error of January 03 in dSTEC-GPS assessment (TECU) | Relative error of January 03 in dSTEC-GPS assessment (%) | RMS error of January 05 in dSTEC-GPS assessment (TECU) | Relative error of January 05 in dSTEC-GPS assessment (%) | Overall standard deviation of the GIM-VTEC versus measured Jason3-VTEC from December 01 of 2020 to January 03 of 2021 in Jason3-VTEC assessment (TECU) | Overall standard deviation of GIM-VTEC versus measured Jason3-VTEC from January 04 of 2021 to March 01 of 2021 in Jason3-VTEC assessment (TECU) |
|---|---|---|---|---|---|---|
| corg | 2.90 | 45.07 | 3.35 | 49.20 | 3.1 | 2.9 |
| ehrg | 2.54 | 39.55 | 2.81 | 41.23 | 3.0 | 2.8 |
| emrg | 2.62 | 40.75 | 2.73 | 40.08 | 3.2 | 2.9 |
| esrg | 2.70 | 41.98 | 3.06 | 44.99 | 3.2 | 3.0 |
| igrg | 2.60 | 40.40 | 3.06 | 44.99 | 2.9 | 2.8 |
| jprg | 2.73 | 42.46 | 2.86 | 41.98 | 2.8 | 2.7 |
| uhrg | 1.91 | 29.69 | 2.21 | 32.43 | 3.9 | 2.8 |
| uprg | 2.04 | 31.80 | 2.41 | 35.39 | 3.9 | 2.8 |
| uqrg | 1.89 | 29.44 | 2.19 | 32.24 | 3.5 | 2.8 |
| whrg | 2.42 | 37.63 | 2.65 | 38.94 | 3.0 | 2.8 |
| igsg | 2.33 | 36.25 | 2.57 | 37.74 | 2.6 | 2.5 |
| crtg | 3.36 | 52.25 | 3.86 | 56.67 | 3.6 | 3.2 |
| crfg | 4.29 | 66.67 | 3.92 | 57.56 | 3.7 | 3.2 |
| cnes | **3.35** | 52.13 | 3.74 | 54.86 | 3.5 | 3.4 |
| cnfs | 3.58 | 55.73 | 4.62 | 67.88 | 3.5 | 3.4 |
| upc1 | 3.85 | 59.91 | **2.80** | 41.06 | 4.3 | **2.7** |
| upf1 | 3.87 | 60.20 | 2.81 | 41.26 | 4.5 | **2.7** |
| whu0 | 5.19 | 80.69 | 5.45 | 79.84 | 4.3 | 4.4 |
| whf0 | 5.31 | 82.61 | 5.54 | 81.28 | 4.3 | 4.4 |
| whu1 | 4.37 | 67.97 | 4.40 | 64.55 | 4.3 | 3.8 |
| irtg | 4.11 | 63.86 | 3.37 | 49.47 | **3.3** | 2.8 |

value in bold font means the corresponding RT-GIM has the best performance among the remaining RT-GIMs in each column, and values of irtg are underlined for comparison.

where $RMS_{\delta,g}$ is the RMS error of GIM $g$. And $\delta_{g,i}$ is the dSTEC error of GIM $g$ similar to Eq. 8, while the reference angle of Eq. 8 is replaced by the maximum elevation angle within a continuous arc. $N_S$ is the number of involved observations. $O_{\Delta_{S_{GPS}}}$ is the dSTEC-GPS observation. $RMS_{\Delta_{S_{GPS}}}$ is the RMS of the observed dSTEC-GPS. $Relative\ error_g$ is the relative error of GIM $g$.

As shown in Table 4, the RMS error of most post-processed GIMs reaches around 2 or 3 TECU, while the RMS error ranges from 2.8 to 5.54 TECU for RT-GIMs. The transition of UPC RT-GIM (upf1) from USRG to UADG is apparent in the dSTEC-GPS assessment, and the RMS error of IGS RT-GIM (irtg) decreased from 4.11 to 3.37 TECU due to the improvement of UPC RT-GIM. After the transition of UPC RT-GIM, the performance of upf1 and irtg is comparable with most post-processed GIMs. Similar to the performance in Jason3-VTEC assessment, the accuracy of remaining RT-GIMs is close to post-processed GIMs. And the RMS error of 2-hour shifted whu1 is around 4.4 TECU which is better than the whu0. Therefore, the 2-hour shift is recommended for $\lambda_{S,t}$ in Eq. 4. It should be pointed out that the performance of RT-GIMs with the full temporal resolution is slightly worse than 20-minute RT-GIMs. Furthermore, the full temporal resolution RT-GIM is even worse than the GIM obtained by linear interpolation of the 20-minute RT-GIM in a sun-fixed reference frame. This is coincident with a smaller number of ionospheric observations at shorter time scales. In Fig. 4, the performance of IGS RT-GIMs after the upgrade of UPC interpolation method in dSTEC-GPS assessment is represented. The higher values of RMS errors occur around the equator and southern hemisphere for all the RT-GIMs. And the higher values might be caused by the high electron density gradients at the equator and the sparse distribution of real-time stations in the southern hemisphere.

## 3.3 The sensibility of real-time weighting technique

RT-dSTEC assessment of RT-GIMs was automatically running in real-time mode, and used for real-time weighting in the combination of IGS RT-GIMs. In order to compare with the dSTEC-GPS assessment, the RT-dSTEC assessment with real-time stations in Fig. 1 was also performed on January 03 and January 05 in 2021. As can be seen in Table 5, the rank of RT-GIMs in RT-dSTEC assessment is similar to dSTEC-GPS assessment, but the RMS error values are larger. And the larger RMS error is coinciding with the much lower elevation angle of the observation reference in RT-dSTEC assessment.

**Table 5.** RMS errors of RT-GIMs in RT-dSTEC assessment on January 03 and January 05 in 2021.

| GIM | RMS error of January 03 (TECU) | RMS error of January 05 (TECU) |
|-----|-------------------------------|-------------------------------|
| upc1 | 4.24 | **3.91** |
| crtg | 4.25 | 4.98 |
| cnes | **3.98** | 4.07 |
| whu0 | 5.94 | 5.81 |

value in bold font means the corresponding RT-GIM has the best performance among the remaining RT-GIMs in each column.

The real-time weights of RT-GIMs can be defined as the normalized inverse of the squared RMS of RT-dSTEC errors and

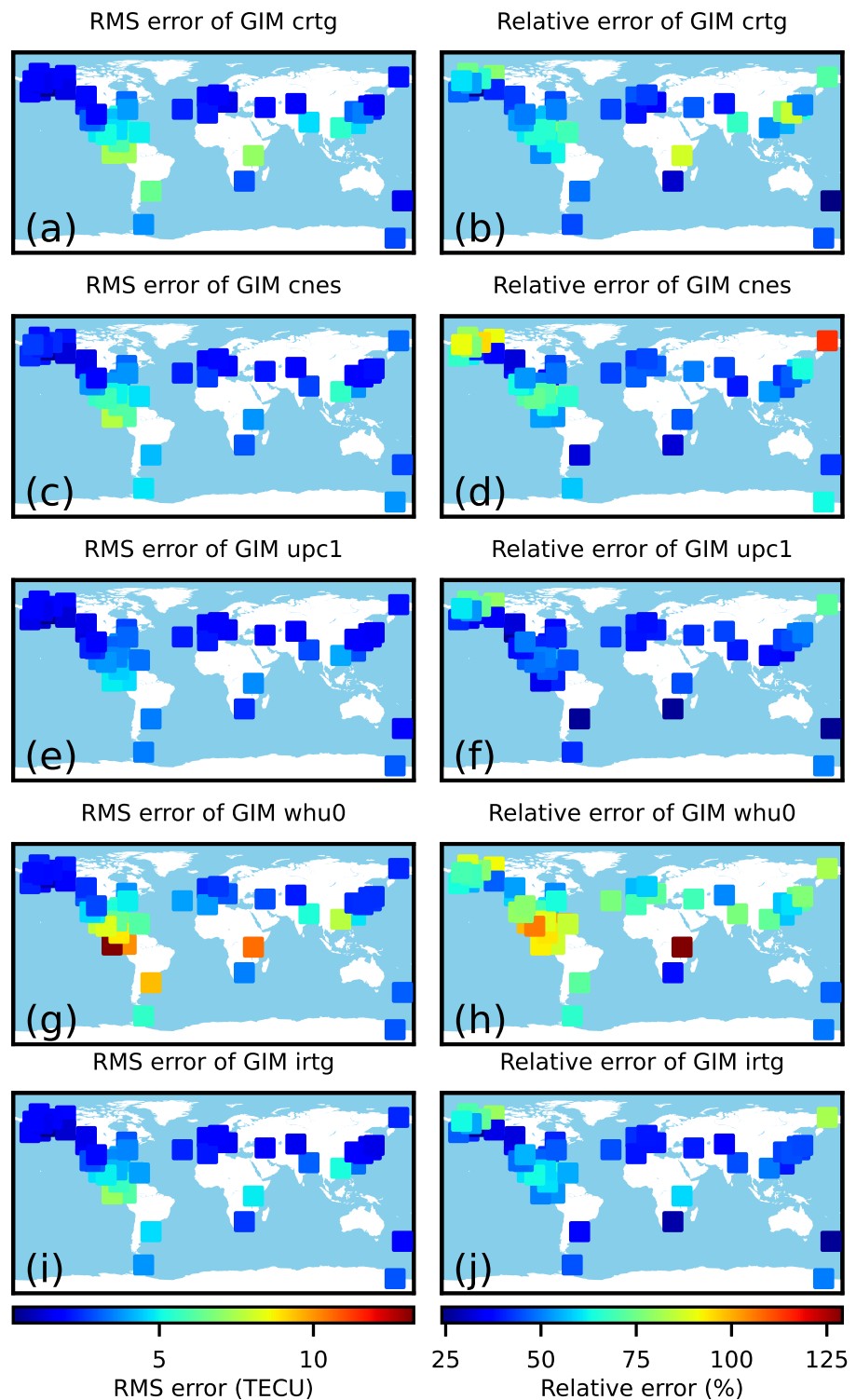

**Figure 4.** The distribution of dSTEC-GPS results on January 05 of 2021 (after the improvement of UPC interpolation technique).

represent the accuracy of RT-GIMs in the RT-dSTEC assessment. For each RT-GIM, the number of daily winning epochs is computed by counting the number of epochs within the day when the one RT-GIM is better than the other RT-GIMs. The evolution of daily winning epochs of RT-GIMs shown in the bottom figure of Fig. 5 is consistent with the Jason3-VTEC assessment. The upc1 was not involved in the combination from December 15 of 2020 to January 2 of 2021 when the dissemination of upc1 was stopped, as can be seen in the bottom figure of Fig. 5. The significant improvement of the transition of upc1 from USRG to UADG shown in dSTEC-GPS and Jason3-VTEC assessment is also obvious in the top figure of Fig. 5. In addition, the daily winning epochs evolution and the transition in Fig. 5 are consistent with the accuracy of RT-GIMs providing a combined RT-GIM which is one of the best RT-GIMs, as shown in the altimeter-based and dSTEC-based assessments. The good performance of the combination algorithm can be mainly explained from the point of view of the weights, i.e. the sensitivity of the dSTEC error to the quality of the RT-GIMs. But also from the point of view of the linear combination that can play a positive role under any potential negative correlation between the performance of pairs of involved RT-GIMs.

### 3.4 The response of RT-GIMs to recent minor geomagnetic storms

The Global Electron Content (GEC) is defined as the total number of free electrons in the ionosphere. Hence the GEC can be estimated from the summation of the product of the VTEC value and the area of the corresponding GIM cell. In addition, GEC has been used as ionospheric index (Afraimovich et al., 2006; Hernández-Pajares et al., 2009). With the purpose of further checking the consistency of IGS RT-GIMs, the GEC of RT-GIMs was calculated and compared from January 24 to January 29 in 2021. It should be noted that the solar activity is low in January of 2021. During the selected period, several weak geomagnetic storms and one moderate geomagnetic storm occurred according to the classification of geomagnetic indices (Loewe and Prölss, 1997; Gonzalez et al., 1999), and the GEC evolution can be seen in Fig. 6. The GEC of CNES RT-GIM (cnfs) is slightly different from other RT-GIMs, and seems to be caused by the bias in CNES RT-GIM. There are some jumps in the GEC evolution of CAS RT-GIM (crfg) and WHU RT-GIM (whf0), and the jumps might be related to the handling of day boundary or unreal predicted GIM in certain cases. Compared with IGS final combined GIM (igsg), the good performance of global VTEC representation with upf1 and irfg can be seen in Fig. 6. In addition, the response of upf1 and irfg to the recent minor geomagnetic storms (detected by 3-hour ap and 1-hour Dst indices) is apparent and also similar to the post-processed IGS final combined GIM (igsg).

### 4 Data availability

The IGS real-time combined GIMs during the testing period are available from Zenodo at http://doi.org/10.5281/zenodo.5042622 (Liu et al., 2021b) in IONEX format (Schaer et al., 1998). In addition, more archived IGS combined RT-GIMs can be found at http://chapman.upc.es/irtg/archive/ and the latest IGS combined RT-GIMs are available in real-time mode at http://chapman.upc.es/irtg/last_results/.

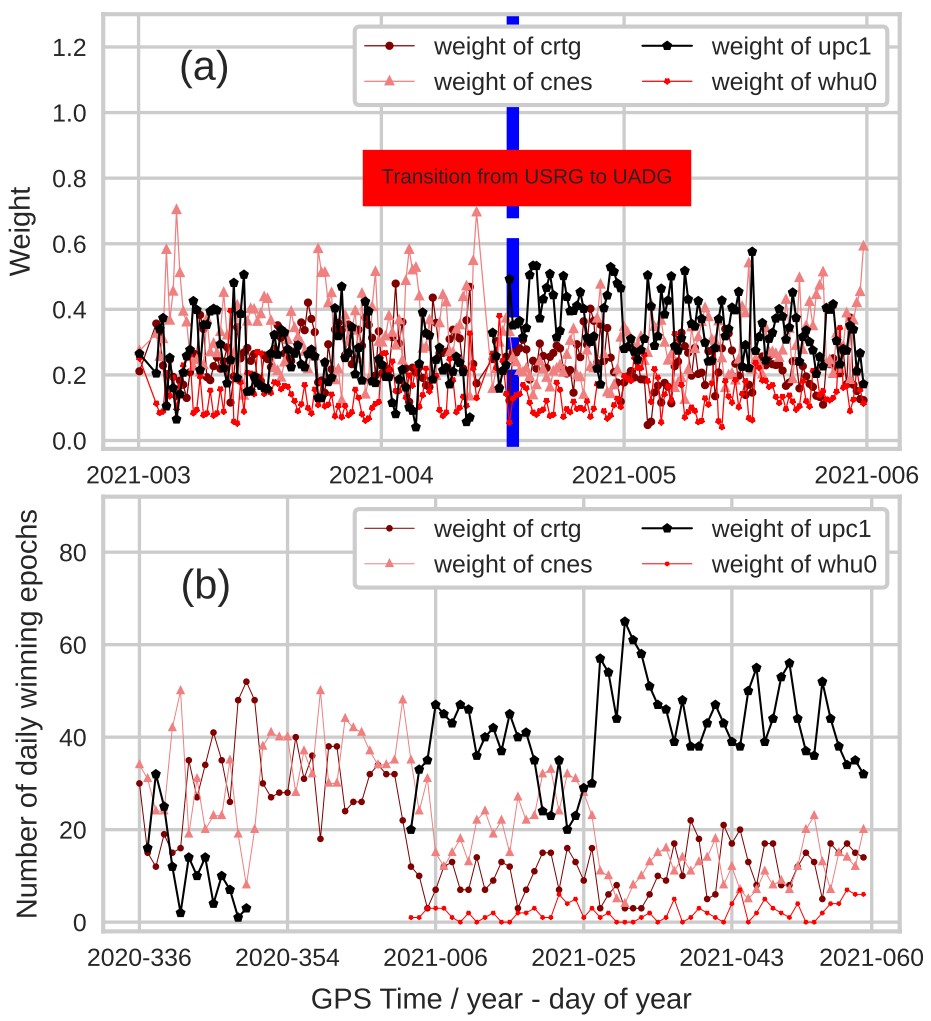

**Figure 5.** The evolution of real-time weights and daily winning epochs of RT-GIMs. (a) The real-time weights from January 03 of 2021 to January 05 of 2021. (b) The daily number of epochs when one of the RT-GIMs is better than the others from December 01 of 2020 to March 01 of 2021.

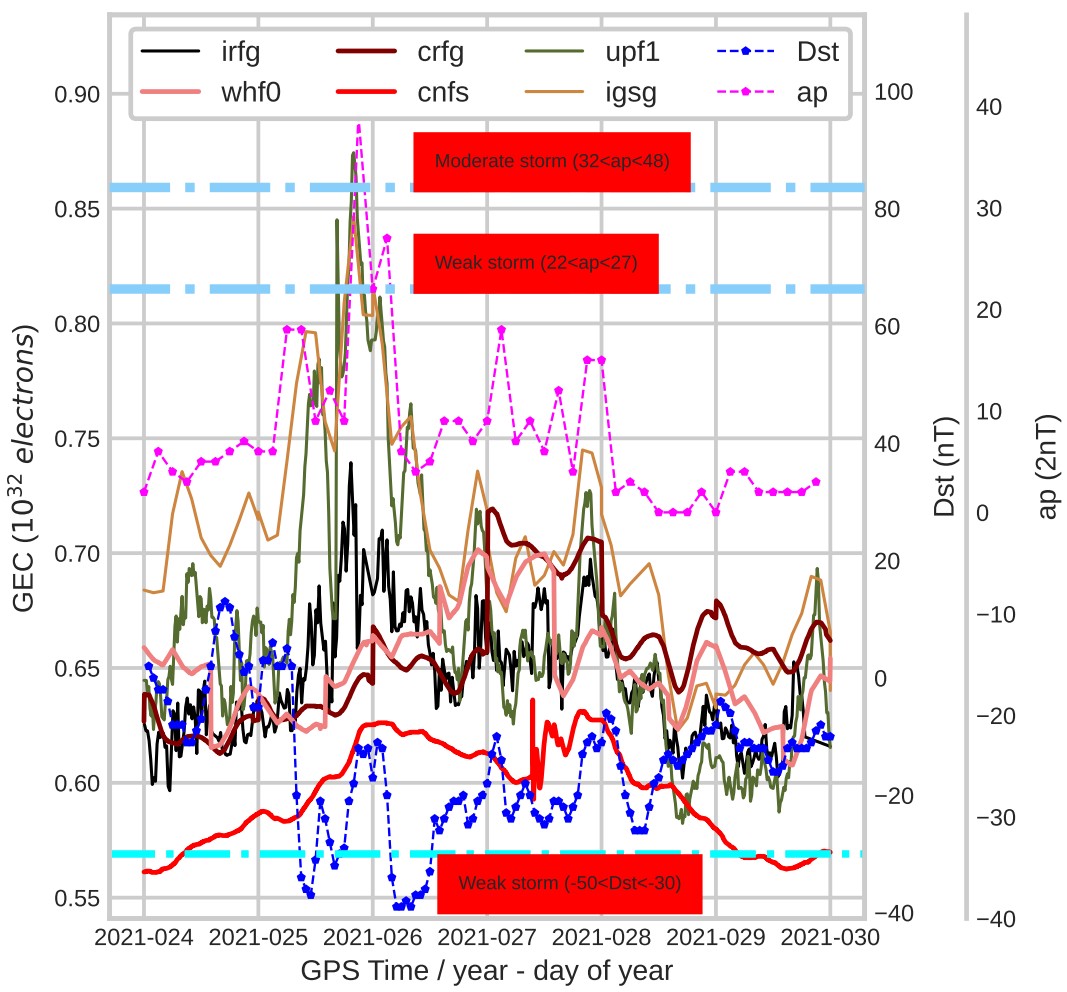

**Figure 6.** The GEC, ap and Dst evolution of RT-GIMs from January 24 to January 29 of 2021 during the low solar activity period.

## 5 Conclusions

In this paper, we have summarized the computation methods of RT-GIMs from four individual IGS ionosphere centers and introduced the new version of IGS combined RT-GIM. According to the results of Jason3-VTEC and dSTEC-GPS assessment, it could be concluded as follows:

- The real-time weighting technique for the generation of IGS combined RT-GIM is performing well when it is compared with Jason3-VTEC and dSTEC-GPS assessment.

- The transition of UPC RT-GIM from USRG to UADG is obvious in all involved assessments and also demonstrates the sensibility of the real-time weighting technique to RT-GIMs when the accuracy of RT-GIMs is increased.

– The quality of most IGS RT-GIMs is close to post-processed GIMs.

    – The difference among RT-GIMs with 20-minute and full temporal resolution can be neglected over oceans in Jason3-VTEC assessment (see Fig. 3 and Table 4), while the difference is visible in some RT-GIMs over continental regions in dSTEC-GPS assessment (see Table 4). The lower accuracy of GIMs with full temporal resolution (2 or 5 minutes) might be related to the uneven distribution of ionospheric observations, the weight between predicted GIMs and real-

time observations. Combined with the previous study (Liu et al., 2021a), it is suggested to find a more suitable temporal resolution for the generation of RT-GIM in sun-fixed reference frame.

In addition, the GEC evolution of UPC RT-GIM and IGS combined RT-GIM is close to the GEC evolution of IGS final combined GIM in post-processing mode, and has an obvious response to the geomagnetic storm during the low solar activity period. Future improvements might include:

– To broadcast real-time RMS maps that can be useful for the positioning users.

    – To increase the accuracy of high temporal resolution RT-GIMs. In addition, higher maximum spherical harmonic degrees might be adopted to increase the accuracy and spatial resolution of RT-GIMs.

    – Coinciding with a much larger number of RT-GNSS receivers in the future, the dSTEC weighting might be improved by replacing the "internal" by the "external" receivers, i.e. not used by any real-time analysis centers. In this way the

weighting would be sensitive as well to the interpolation/extrapolation error of the different real-time ionospheric GIMs to be combined. And the resulting combination might behave better.

    – To increase the number of worldwide GNSS receivers used for the RT-dSTEC up to more than 100. In this way we will be able to study the potential upgrade of the present global weighting to a regional weighting among other potential improvements in the combination strategy.

*Author contributions.* QL wrote the manuscript. QL developed the updated combination software with contributions from DRD, HY and MHP. QL and MHP designed the research, with contributions from HY, EMM, DRD and AGR. QL, HY, EMM, MHP, ZL, NW, DL, AB, Q. Zhao and Q. Zhang provided the real-time GIMs of the corresponding IGS centers. AH, MS, GW and AS contributed in creating the framework of the real-time IGS service, the ionospheric message format and BNC open software updates. LA suggested the initial idea of this work. AK, SS, JF, AK, RGF and AGR contributed in the generation of rapid and final IGS GIMs used as additional reference in the

manuscript.

*Competing interests.* The authors declare that they have not conflict of interest.

*Acknowledgements.* The first author is grateful for the financial support of the China Scholarship Council (CSC). The contribution from UPC-IonSAT authors was partially supported by the European Union funded project 101007599 - PITHIA-NRF and by the ESSP/ICAO funded project TEC4SpaW. The work of AK is supported by the National Centre for Research and Development, Poland, through grant ARTEMIS (decision numbers DWM/PL-CHN/97/2019 and WPC1/ARTEMIS/2019). The authors are thankful to the collaborative and friendly framework of the International GNSS Service, an organization providing first-class open data and open products (Johnston et al., 2017). The VTEC data from Jason3-altimeter were gathered from the NASA EOSDIS Physical Oceanography Distributed Active Archive Center (PO.DAAC) at the Jet Propulsion Laboratory, Pasadena, CA (http://dx.doi.org/10.5067/GHGMR-4FJ01) and the National Oceanic and Atmospheric Administration (NOAA). We are also thankful to the ap and Dst indexs from GeoForschungsZentrum (GFZ) and World Data Center (WDC) for Geomagnetism, Kyoto.

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
