# Peer review of "The cooperative IGS RT-GIMs: a reliable estimation of the global ionospheric electron content distribution in real-time"

_Earth System Science Data, 2021_

## Author Response (AR1)

Manuscript Title: The cooperative IGS RT-GIMs: a global and accurate estimation of the ionospheric electron content distribution in real-time

Manuscript Number: essd-2021-136

Corresponding Author: Manuel Hernández-Pajares

Dear editor, dear anonymous referees,

We thank you for constructive comments and suggestions. They can be very helpful to improve the quality of manuscript and also to increase the readability.

We will firstly provide a point-by-point response to editor and two reviewers' comments in red font. Then the track of the changes and also the revised version of manuscript will be attached and named as "diff.pdf" and "IRTG-ESSD-v3.pdf", respectively. In addition a typo has just been fixed in the first raw of Table 4 (dSTEC error of CORG GIM).

Thank you.

Please note that reviewers' comments are in italics while our answers are not.

Yours sincerely, on behalf of the co-authors,

Qi Liu and Manuel Hernández-Pajares

**Answers to Editor Christian Voigt, essd-2021-136-EC1.pdf**

**Comment E1.1** *Dear Authors,*

*Thank you very much for your detailed response to all referee comments. I have no additonal comments or recommendations. We are now asking you to submit a revised version of your manuscript.*

*Best regards,*

*Christian Voigt*

**Answer to E1.1** Many thanks for your comments to the manuscript. Best regards.

**Answers to Anonymous Referee 1, essd-2021-136-RC1.pdf**

**Comment R1.1** *This paper describes the four real-time GIM products by CAS, CNES, UPC, and WHU, as well as the real-time IGS GIM combination. The results are analyzed for one month at the beginning of 2021 by means of a comparison with Jason3 VTEC data and by the differential STEC technique. In particular, the paper focuses on the update of the UPC solution and its impact on the combined IGS solution. The topic is highly relevant, and the new UPC and IGS solutions justify publishing this contribution. The GIMs of the combined IGS solution are provided in the commonly used and well documented IONEX format and are therefore easily usable by the community. The data set including only one month is a bit limited, especially considering that it starts only one day before the switch of the UPC solution on January 4. In my opinion, the content of the manuscript is worth publishing, but the presentation requires substantial improvement concerning language, clarity of presentation, and consistency of mathematical expressions and equations, for which examples are given below.*

**Answer to R1.1** We thank the reviewer for the efficient and professional editing of our manuscript. We have tried to summarize as well in this manuscript the computation methods of IGS RT-GIMs and the accuracy of different real-time GIMs in Jason3-VTEC assessment and dSTEC assessment, and also validate the effectiveness of real-time weighting technique for combination, the response of RT-GIM to geomagnetic storm, and the influence of temporal resolution on RT-GIM.

Following your suggestion, now we have extended the experiment. We add data one month before January 2021 and one month after January 2021 (from December 01 of 2020 to March 01 of 2021). The new results are similar and consistent with previous experiment. Fig. 3, Fig. 5. and Table 4 have been reproduced and the corresponding descriptions have been changed. In particular, the overall standard deviation of upc1 VTEC versus measured Jason3-VTEC drops from 4.3 to 2.7 TECU and, in agreement with that, the standard deviation of irtg VTEC versus measured Jason3-VTEC decreases from 3.3 to 2.8 TECU. Accordingly, the description of the figures and tables has also been modified. As shown and explained in Eq. 4, the whu0 is shifted by 0 hours. To see the influence of phase-shifted $\lambda_{S,t}$, the whu0 is manually shifted by 2 hours (i.e., take $t_0$ as 2 hours for whu0 in Eq.4) in post-processing mode. As can be seen in Fig. 3 and Table 4, the 2-hour shifted WHU RT-GIM (whu1) is slightly better than whu0.

**Comment R1.2** *The mathematical expressions throughout the entire paper are very inconsistent, incomplete, and also incorrect, e.g.*

**Answer to R1.2** Sorry for the misleading and many thanks for your kind suggestions. Because the article joins the products of different centers, we have preserved the notation of the original contributions in order to facilitate the references to the original articles in the first version of manuscript.

Now all the equations are reorganized and the symbols are unified.

**Comment R1.3** *The parameters should have consistent indexes, e.g., in (1) P, L, and STEC have no indexes in the first two lines, but then P and L have a time index k in the last line (whereas STEC and Phi do not). The biases, however, have a satellite and receiver index. Later in (4), the STEC has indexes for r and s, and depends on latitude, longitude, time, and zenith angle (which doesn't make sense by itself, as receiver+satellite and time uniquely define the STEC parameter, the other dependencies are superfluous). In (7), the observations and VTEC values explicitly depend on time via ...(t), which is again a different representation compared to a subscript.*

**Answer to R1.3** Now equation (1) is split into three equations. k is the length of smoothing arc from beginning epoch to epoch t. And the time index t is added at each necessary variable. The unnecessary indexes in equation (4) are remove. The time indexes in equation (7) are unified as t in subscript.

**Comment R1.4** *(2)is just a copy of the second line of (1), so why is it repeated?*
**Answer to R1.4** Yes. The initial intention of equation (2) is to represent the LI in a separate equation to avoid confusion. But it seems not very clear. Now the previous equation (1) is split into three equations to avoid duplication.

**Comment R1.5** *c is used as the speed of light in L86, but in (4) it is a different constant that is not defined.*
**Answer to R1.5** Sorry for this contradictory definition. The definition of c has been modified.

**Comment R1.6** *The symbol for the VTEC is mostly VTEC, but sometimes V (5,7).*
**Answer to R1.6** Now all the symbols of VTEC are unified.

**Comment R1.7** *In (6) in the last line the authors summarize over i, which means the result cannot depend on i, so this equation cannot be correct.*
**Answer to R1.7** Sorry for the misleading. Now the equation is modified.

**Comment R1.8** *In (7), the VTEC depends on t, so delta should also depend on t.*
**Answer to R1.8** Yes, you're right. Now this has been modified.

**Comment R1.9** *N is used in the paper as the number of ACs, the number Jason3 VTEC points during one day, the number of dSTEC points, and the maximum degree of the spherical harmonics, which should be avoided to facilitate the readability.*
**Answer to R1.9** Thanks for your suggestion. Now the symbol of N are defined in different forms to avoid misleading.

**Comment R1.10** *Why is the phase wind-up term Phi introduced in (1) and (2), when it is simply ignored afterwards? This is an a-priori correction similar to antenna PCVs and not a parameter, and should therefore not be listed, or at least it should be explained how it is treated.*
**Answer to R1.10** Yes, the phase wind-up is defined in Equation (1) and (2). Later in remaining equations, we only use VTEC extracted from Equation (1) and (2). Now we removed the wind-up item in the equation and noted the wind-up in the description.

**Comment R1.11** *Taking the SQUARE of the ROOT-mean-square in (6) is a bit pointless and simply is the mean-square.*
**Answer to R1.11** Sorry for the misleading. Now we replace "$t_i$ is the inverse of the squared RMS of RT-dSTEC error $RMS_{\delta,i}^2$ and $\delta_i$ is the RT-dSTEC error of IGS center $i$ in the RT-dSTEC assessment."
with
"$RMS_{\delta,g,t}$ is the root mean square of RT-dSTEC error. $I_{g,t}$ is the inverse of the mean square of RT-dSTEC error. $N_t$ is the number of RT-dSTEC observations from the beginning epoch to current epoch $t$. $\delta_{g,i}$ is the RT-dSTEC error of RT-GIM $g$ in the RT-dSTEC assessment.".

**Comment R1.12** *The presentation generally is too unclear and unprecise, and should be revised in this regard, e.g.: - L10 'after the transition of interpolation technique': This cannot be understood at this point, as this is only explained later.*

**Answer to R1.12** Thank for your kind suggestion. 'after the transition of interpolation technique' has been replace by "after the improvement of interpolation technique".

**Comment R1.13** *L25 'due to the good performance and global distribution of VTEC': What is the performance of VTEC?*

**Answer to R1.13** Sorry for the misleading, "the good performance" was supposed to mean the high accuracy. Now we replace "Due to the good performance and global distribution of VTEC" with
"Due to the high quality and global distribution of VTEC estimation".

**Comment R1.14** *L57: RTK over medium and long baselines can also achieve centimeter level without any ionospheric information, it might just take longer.*

**Answer to R1.14** Sorry for the misleading. We meant the instantaneous (single-epoch) RTK over medium and long-baseline. For instantaneous (single-epoch) RTK over medium and long-baseline, the ionospheric information is able to increase the success rate and reliability.
Now, we replace "while the Real-Time Kinematic (RTK) Positioning over medium and long-baseline is able to achieve few centimeters level for rover stations"
with
"while the instantaneous (single-epoch) Real-Time Kinematic (RTK) Positioning over medium and long-baseline is able to obtain higher success rate of the ambiguity fixing and reliability for rover stations in few centimeters level"

**Comment R1.15** *It is often stated that the 'STEC is extracted from the GF combination'. This is not correct. This is the STEC biased by ambiguities or DCBs. The unbiased STEC can only be obtained by fitting an ionospheric model to the observations. This should be made more precise.*

**Answer to R1.15** Thank you for your kind suggestion. The original expression might be misleading.
Now we add the sentence "It should be noted that extraction of STEC in an unbiased way can be obtained by fitting an ionospheric model to the observations." at the beginning of Section 2.1 to define the extraction of unbiased STEC.
Then we replace "The geometry-free (GF) combination of pseudrange and carrier phase observations is formed to extract STEC within a continuous arc."
with
" The geometry-free (GF) combination of pseudorange and carrier phase observations is formed to extract STEC and DCB in an unbiased way by fitting an ionospheric model (for example, spherical harmonic model)."

**Comment R1.16** *L96 'Since ... adopts spherical harmonic expansion...': This is not the reason that some ACs use the spherical harmonic model.*

**Answer to R1.16** Sorry for the misleading. Now we replace "Since the dissemination of RT-GIMs adopts spherical harmonic expansions, some ionosphere centers (CAS, CNES, WHU) directly estimate spherical harmonic coefficients in sun-fixed reference frame as Eq. 3"

with
"Some ionosphere centers (CAS, CNES, WHU) directly estimate and disseminate spherical harmonic coefficients in sun-fixed reference frame as Eq. 4".

**Comment R1.17** *The description of the CAS method should be more clear. What I currently understand is that they compute local VTEC models, then transform the VTEC to STEC, then back to VTEC, and then estimate a global model. Is this really the case?*

**Answer to R1.17** Sorry for the misleading. Actually, the local VTEC model is used for the DCB estimation. As we written "The Eq. 1 and Eq. 4 from 130 global stations are combined to calculate real-time line-of-sight STEC, which is then converted to VTEC with an ionospheric mapping function in single-layer assumption at 450 km." in the manuscript, the real-time STEC is computed by $\tilde{P}_{GF,k} - c \cdot (D_r + D^s)$ and the STEC is converted into VTEC by means of mapping function. With the global distributed VTEC, the spherical harmonic model is used for global VTEC modelling. In addition, the extra predicted ionospheric information is introduced, as mentioned in the paper "CAS predicted TEC information is also included for RT-GIM computation".

Maybe the sentences are not very clear. Now we replace "The Eq. 1 and Eq. 4 from 130 global stations are combined to calculate real-time line-of-sight STEC, which is then converted to VTEC with an ionospheric mapping function in single-layer assumption at 450 km. The VTEC is directly modeled in a solar-geographic reference frame as Eq. 3. To mitigate the impacts of the unstable real-time data streams, e.g. the sudden interruption of the data streams, CAS predicted TEC information is also included for RT-GIM computation"
with
"The real-time STEC is computed by subtracting estimated DCB in Eq. 3 from $\tilde{P}_{GF,t}$ in Eq. 1 and the STEC is converted into VTEC by means of mapping function. The real-time VTEC is directly modeled in a solar-geographic reference frame as Eq. 3. To mitigate the impacts of the unstable real-time data streams, e.g. the sudden interruption of the data streams, CAS predicted TEC information is also included for RT-GIM computation. The broadcasted CAS RT-GIM is computed by the weighted combination of real-time spherical harmonic coefficients and predicted ionospheric information."

**Comment R1.18** *I cannot follow the description of the UPC method in L148 and following, maybe this could be presented in a more intuitive way. In (5) it is stated that Gt is the generated RT-GIM. Gt seems to be a vector, but a GIM is a set of grid points characterized with latitude, longitude, time, and VTEC value. Dt is simply defined as 'dictionary matrix'. Either all of this should be explained or simply described in a clear way, since the reference is given.*

**Answer to R1.18** We apologize for the lack of clarity. We have introduced the following modification:
The intuition behind the method is the following, we construct a linear combination of past maps which at the IPP are most similar to our measures. For mathematical convenience, we represent each 2-D GIM as a 1-D vector (I.e., the columns are stacked along the meridian in order to create a vector of all the grid points of the map). This is justified because the measure of similarity is done over cells of 2.5×5.0 degrees in the maps, and therefore the underlying $\mathbb{R}^2$ (coordinate space of dimension 2) structure is not relevant for computing euclidean distances in $\ell_2$ norm. The method for obtaining the coefficients of the linear combination of past maps is based on equations (6), which were first introduced in the problem of face recognition. The problem of face recognition has in common with the current problem that there are occlusions in the face image/ionospheric

map, that is the regions not covered by stations or the effect of glasses. These effects have to be taken into account when selecting the past maps, and should not introduce a bias and this is done by using the double norm criterion of equation (6).

[revised manuscript text omitted]

By the way, more details can be see in the origin manuscript and it is now available online `https://doi.org/10.1007/s00190-021-01525-5`.

**Comment R1.19** *Figure 5 is difficult to read, maybe a smoothed representation would help?*
**Answer to R1.19** In order to smooth the Figure 5, we firstly change the zoom period in the top Figure 5(a). Now only real-time weights during 4 days are available to focus on the weight transition from USRG to UADG. Then we try to plot the daily winning epochs of RT-GIMs in Figure 5(b) from December 01 of 2020 to March01 of 2021 to avoid the noisy evolution of weight in long-time period. For each RT-GIM, the number of daily winning epochs is computed by counting the number of epochs within the day when the one RT-GIM is better than the other RT-GIMs.

**Comment R1.20** *L296 'RT-weighting ... turns out to be effective': What does this mean? Effective for what?*
**Answer to R1.20** Sorry for the ambiguous expression. Now we replace "The RT-weighting technique for the generation of IGS combined RT-GIM turns out to be effective"
with
"The RT-weighting technique for the generation of IGS combined RT-GIM is performing well"

**Comment R1.21** *The language of the paper as well as some technical aspects should be carefully revised, e.g., missing articles should be added, sentences that are only equations (L86) should be avoided, and abbreviations have to be defined at first use (GIM in L10, but used in L3, VTEC is used in L23, but TEC is defined in L26, SHE is defined in L166 but only used within this paragraph, outside the full expression is used,...).*
**Answer to R1.21** Many thanks for your kind suggestions. We have just updated the reference articles including the missing articles and also a recent accepted article about UADG.
Now the sentences that are only equations has been modified.
In L3, we use RT-GIM and RT-GIM is defined in L3. Following the requirement of journal "Abbreviations need to be defined in the abstract and then again at the first instance in the rest of the text.", we define abbreviation the first time when we use it in a sentence at the abstract and main text. Now all the abbreviations are reorganized and redefined.

**Answers to Anonymous Referee 2, essd-2021-136-RC2.pdf**

**Comment R2.1** *The authors address in their study the generation of a specific real-time data product of IGS. This is the IGS combined Real-Time Global Ionosphere Map (RT-GIM) generated by real-time weighting of RT-GIMs computed simultaneously at four IGS real-time ionosphere centers. Because different centers use different approaches to estimate global TEC the combination of different approaches is not trivial. Consequently, the authors discuss is topic in detail. Validation of the performance of contributing ionosphere center related TEC estimates and the combined products is carried out by using independent altimeter data (1 month) from Jason-3 satellite over oceans and the dSTEC technique over continents (2 days). Comparison is also made with better conditioned post-processed GIMs. The authors finally conclude that the IGS RT-GIMs are a reliable source of real-time global VTEC information having a great potential for real-time GNSS applications.*
*The results should principally be of interest for readers of ESSD.*
*The manuscript is well organized. Nevertheless, there are a few points which need improvement/clarification in a revised version as indicated in the subsequent comments:*

**Answer to R2.1** Thanks for your efficient and professional editing and kind suggestions. We have modified the manuscript and answered your questions point-by-point as following.

**Comment R2.2** *Science*
*Considering the Jason3-VTEC assessment the constant bias estimate includes practically the plasmaspheric electron content above 1300 km height which definitively not constant on global scale. This is clearly a weak point in the subsequent weighting practice which is based on the RMS error between VTEC (Jason) and the GIMs. This critical point concerns also the dSTEC technique if arc lengths between measurement and reference point are large, i.e. if the ray path geometry changes significantly. Additionally, mapping function errors are also included in the RMS error that is used as weighting criterium for different centers. Here arises also the question whether the different centers use exactly the same data base for the construction of their GIMs. If not, there is another source of uncertainty for estimating the weighting of different centers.*
*I think the authors should discuss these problems in their manuscript adequately.*

**Answer to R2.2** Thanks for your kind suggestions. We are focusing on the standard deviation of the difference between GIM-VTEC and Jason3-VTEC to avoid the Jason3-altimeter bias and the mean bias component of the plasmaspheric electron content in the assessment. The plasmaspheric electron content variation is up to a few TECU and is relatively a small part when compared with the GIM errors over the oceans. As a consequence, the GIM validation based on dual-frequency Jason3-altimeter measurements is sensitive to the actual error of the GIMs on the oceans where are the most challenging regions for GIMs (containing few nearby receivers in such regions) and typically far from permanent GNSS receivers potentially contributing to the GIM (see details in [1, 2]). As summarized in section 2.3, the weighting of RT-GIMs to generate the IGS combined RT-GIM is exclusively based on RT-dSTEC weighting technique and use the GF combination of phase-only observations to calculate the RMS of dSTEC error by Eq. 8.
The dSTEC observations provide the direct measurements of the difference of STEC within a continuous phase-arc involving different geometries but avoiding the huge mapping function errors by applying an elevation mask of 15 degrees. And the mapping function is used by GNSS users to convert GIM-VTEC to GIM-STEC for GNSS positioning. Therefore, the dSTEC observations, containing different geometries and mapping function values, are accurate and direct measurements for evaluating ray path GIM-STEC which is commonly used by GNSS users to calculate

ionospheric correction. In addition, the common agreed ionospheric thin layer model is set to be 450 km height in the generation of GIM to provide VTEC in a consistent way for different ionospheric analysis centers. And in this way the GNSS users are able to consistently recover a most accurate STEC from GIM-VTEC by the commonly agreed mapping function.

The GIM error versus JASON-VTEC measurements have a high correlation with the GIM error versus dSTEC-GPS based measurements, although the JASON-VTEC measurements are vertical and the dSTEC-GPS measurements are slant. As demonstrated in [2], the Jason3-VTEC assessment and dSTEC-GPS assessment are independent and consistent for GIM evaluation. In other words, the slant ray path geometry changes does not affect the capability of dSTEC reference data to rank the GIM, and the plasmaspheric component does not significantly affect the assessment of GIMs based on Jason3-VTEC data.

In the manuscript, we have made some modifications for explanation:
a) We replace "The VTEC from the Jason3-altimeter was gathered as an external reference over oceans which were also the most challenging regions for GIMs (typically containing few nearby receivers in such regions)." with "The VTEC from the Jason3-altimeter was gathered as an external reference over the oceans."
b) We replace "Although the electron content above Jason3-altimeter (about 1300 km) is not available and the altimeter bias is around a few TECU, Jason3-VTEC has been proven to be a reliable reference of VTEC on a global scale (Hernández-Pajares et al., 2017). In this context, the daily standard deviation of the difference between Jason3-VTEC and GIM-VTEC was selected as the statistic for GIM assessment in Eq. 9" with "Although the electron content above Jason3-altimeter (about 1300 km) is not available and the altimeter bias is around a few TECU, the standard deviation of the difference between GIM-VTEC and Jason3-VTEC is adopted to avoid the Jason3-altimeter bias and the constant bias component of the plasmasphere in the assessment. The plasmaspheric electron content variation is up to a few TECU and is relatively a small part when compared with the GIM errors over the oceans. And Jason3-VTEC has been proven to be a reliable reference of VTEC on the oceans where are the most challenging regions for GIMs (containing few nearby receivers in such regions) and typically far from permanent GNSS receivers potentially contributing to the GIM (Hernández-Pajares et al., 2017). In this context, the daily standard deviation of the difference between Jason3-VTEC and GIM-VTEC was suitable as the statistic for GIM assessment in Eq. 9"
c) After sentence "In the dSTEC-GPS assessment, the maximum elevation angle within a continuous arc was regarded as the reference angle in Eq. 8.", we add "The dSTEC observations provide the direct measurements of the difference of STEC within a continuous phase-arc involving different geometries. And the mapping function is used by GNSS users to convert GIM-VTEC to GIM-STEC for GNSS positioning. Therefore, the dSTEC observations, containing different geometries and mapping function error, are accurate and direct measurements for evaluating ray path GIM-STEC which is commonly used by GNSS users to calculate ionospheric correction. In addition, the common agreed ionospheric thin layer model is set to be 450 km height in the generation of GIM to provide VTEC in a consistent way for different ionospheric analysis centers. And in this way the GNSS users are able to consistently recover a most accurate STEC from GIM-VTEC by the commonly agreed mapping function."
d) After the sentence "The influence of temporal resolution on RT-GIMs was also shown in this section.", we add "Before detailing the JASON3-VTEC and GPS-dSTEC assessment, it should be taken into account that the GIM error versus JASON-VTEC measurements have a high correlation with the GIM error versus dSTEC-GPS based measurements, although the JASON-VTEC measurements are vertical and the dSTEC-GPS measurements are slant. As demonstrated in [2], the Jason3-VTEC assessment and dSTEC-GPS assessment are independent and consistent for GIM evaluation. In other words, the slant ray path geometry changes does not affect the capability of dSTEC reference data to rank the GIM, and the electron content between the Jason3-altimeter and the GNSS satellites does not significantly affect the assessment of GIMs based on Jason3-VTEC data."

Regarding the point on whether different centers use exactly the same data base for the construction of their GIMs or not: In our knowledge all the IGS ionospheric analysis centers use GNSS data from permanent GNSS receivers. The number of constellations in GNSS data and the distribution of GNSS receivers used by the different analysis centers are not identical (some centers are still using GPS-only data in their official product, while others are using multi-GNSS). But this is not an issue and the use of different techniques for modelling ionosphere VTEC, with complementing benefits, is one fundamental aspect to explain the good behaviour of the combined GIM.

**Comment R2.3** *Data set*
*The data set includes 1 month of Jason 3 vertical TEC data over oceans and 2 days of ground based GNSS data over land. Thus, the data base is very limited to derive general conclusions on physical relationships concerning the physics of the ionosphere. However, the authors use the data set to demonstrate the estimation of VTEC at 4 data different data centers and the procedure of combining their VTEC estimates in near real time. Thus, the data set is appropriate and of high quality. The question is, whether all enters use the same data set to ensure a fair comparison.*

**Answer to R2.3** Thanks for your understanding. Following your and also the first reviewer's suggestion, now we have extended the experiment. We add data one month before January 2021 and one month after January 2021 (from December 01 of 2020 to March 01 of 2021). The new results are similar and consistent with previous experiment. Fig. 3, Fig. 5. and Table 4 have been reproduced and the corresponding descriptions have been changed. In particular, the overall standard deviation of upc1 VTEC versus measured Jason3-VTEC drops from 4.3 to 2.7 TECU and, in agreement with that, the standard deviation of irtg VTEC versus measured Jason3-VTEC decreases from 3.3 to 2.8 TECU. Accordingly, the description of the figures and tables has also been modified. As shown and explained in Eq. 4, the whu0 is shifted by 0 hours. To see the influence of phase-shifted $\lambda_{S,t}$, the whu0 is manually shifted by 2 hours (i.e., take $t_0$ as 2 hours for whu0 in Eq.4) in post-processing mode. As can be seen in Fig. 3 and Table 4, the 2-hour shifted WHU RT-GIM (whu1) is slightly better than whu0.
The answer to "whether all enters use the same data set to ensure a fair comparison." has been included in the last paragraph of "Answer to R2.2".

**Comment R2.4** *Wording*
*Headline: The authors should avoid the term "accurate" in the headline because this requires a clear definition what accuracy means. The authors themselves conclude later in line 311 that the accuracy should be increased.*
**Answer to R2.4** Thank you for your suggestion. Now we replace "The cooperative IGS RT-GIMs: a global and precise estimation of the ionospheric electron content distribution in real-time" with

"The cooperative IGS RT-GIMs: a reliable estmation of the global ionospheric electron content distribution in real-time" in the headline to avoid ambiguity. Actually there is always room for improvement. And that's why we said "To increase the accuracy" in line 311 for the "Future improvements might include:".

**Comment R2.5** *Abstract: The abstract should have clear and compact statements concerning the results of the paper. Thus, for instance, there is a very long sentence covering lines 15- 19 that contains several illustrations in brackets which should be avoided in the abstract.*
**Answer to R2.5** Thank you for your kind suggestion. We have deleted illustrations in brackets.

**Comment R2.6** *Line 8: the real-time weighting technique is sensitive to the accuracy of RT-GIMs As I understand the weighting is dependent from the accuracy, not the technique*
**Answer to R2.6** Sorry for the misleading. We have deleted the "technique" in this sentence.

**Comment R2.7** *Line 203: correct "... IGS-SSR is compatible with RTCM-SSR contents, while IGS-SSR..."*
**Answer to R2.7** Sorry for the misleading. Now we replace "The content of ISG-SSR is compatible with RTCM-SSR contents, while ISG-SSR supports more extensions." with "The content of ISG-SSR is compatible with RTCM-SSR contents. And the IGS-SSR format can support more extensions such as satellite attitude, phase center offsets and variations in the near future."

**Comment R2.8** *Equations*
*(6): Please check the correctness, eq. is not understandable*
**Answer to R2.8** Sorry for the misleading. We have reorganized the equations and the symbols.

**Comment R2.9** *Figures*
*Fig 5: needs precise description, the zoom refers*
**Answer to R2.9** Follow your and the first reviewer's suggestion, we firstly change the zoom period in the top Figure 5(a). Now only real-time weights during 4 days are available to focus on the weight transition from USRG to UADG. The significant improvement of the transition of upc1 from USRG to UADG shown in dSTEC-GPS and Jason3-VTEC assessment is also obvious in the top figure of Fig. 5(a). Then we try to plot the daily winning epochs of RT-GIMs in Figure 5(b) from December 01 of 2020 to March 01 of 2021 to avoid the noisy evolution of weight in long-time period. For each RT-GIM, the number of daily winning epochs is computed by counting the number of epochs within the day when the one RT-GIM is better than the other RT-GIMs.

---

## Author Response (AR2)

Manuscript Title: The cooperative IGS RT-GIMs: a reliable estimation of the global ionospheric electron content distribution in real-time

Manuscript Number: essd-2021-136

Corresponding Author: Manuel Hernández-Pajares

Dear editor, dear anonymous referees,

We thank you for constructive comments and suggestions. They can be very helpful to improve the quality of manuscript and also to increase the readability.

We will provide a point-by-point response to your kind comments in red font.

Thank you.

Please note that reviewers' comments are in italics while our answers are not.

Yours sincerely, on behalf of the co-authors,

Qi Liu and Manuel Hernández-Pajares

**Answers to Editor Christian Voigt,**

**Comment E1.1** *Thank you for your through revision of the manuscript. You will see that Referee 1 agrees with your modifications adressing his/her concerns. Nevertheless, a few mistakes appear in the revised document that need to be corrected.*

**Answer to E1.1** Many thanks for your professional editing of our manuscript. We have just made a modified version of manuscript according to the suggestions.

**Answers to Anonymous Referee 1,**

**Comment R1.1** *I would like to thank the authors for addressing my concerns. With the longer time span presented in the revision, the switch of the UPC interpolation strategy can be seen much clearer. The presented equations are now consistent. I only have a few very small comments*

**Answer to R1.1** Many thanks for your professional editing of our manuscript. We have just made a modified version of manuscript according to your kind suggestions.

**Comment R1.2** *End of Eq. 3: There is a ')' too much.*

**Answer to R1.2** Thank you. This is corrected.

**Comment R1.3** *L126: DCB has already been defined in L95.*

**Answer to R1.3** Yes. Sorry, we forget this. Now it is corrected.

**Comment R1.4** *L262: 'N' should be '$N_J$'*

**Answer to R1.4** Thank you. This is modified.

**Comment R1.5** *L256 and L292: The two sentences starting with 'And' should be reformulated, as they don't seem to be full sentences to me.*

**Answer to R1.5** Thanks for your kind suggestion.

Now the L256 sentence "And Jason3-VTEC has been proven to be a reliable reference of VTEC on the oceans where are the most challenging regions for GIMs (containing few nearby receivers in such regions) and typically far from permanent GNSS receivers potentially contributing to the GIM"

has been replaced by

"Jason3-VTEC has been proven to be a reliable reference of VTEC over the oceans. The oceans are the most challenging regions for GIMs where permanent GNSS receivers are typically far away".

The L292 sentence "And the mapping function is used by GNSS users to convert GIM-VTEC to GIM-STEC for GNSS positioning. Therefore, the dSTEC observations, containing different geometries and mapping function error, are accurate and direct measurements for evaluating ray path GIM-STEC which is commonly used by GNSS users to calculate ionospheric correction."

has been replaced by

"As it has been introduced before, the STEC is proportional to VTEC by means of the ionospheric mapping function. Therefore, the dSTEC error observations (see Eq. 8), containing different geometries and mapping function error, are direct measurements for evaluating GIM-STEC which is commonly used by GNSS users to calculate ionospheric correction."

**Comment R1.6** *Eq. 10: 'O...' needs to have an index 't' and 'i' in the second and third line, respectively (otherwise also the sum in the third line wouldn't make sense).*

**Answer to R1.6** You're right. Thank you. This is modified.

**Comment R1.7** *Fig. 5b: In the legend it should not say 'weight of ...', as the figure represents a number of epochs, not a weight.*

**Answer to R1.7** Thanks for your kind suggestion. Now the legend of Figure 5b has been changed.